# Reusing Pretrained Models by Multi-linear Operators for Efficient Training

**Yu Pan**[♠][∗], **Ye Yuan**[◇], **Yichun Yin**[♡], **Zenglin Xu**[♠][♣][†], **Lifeng Shang**[♡], **Xin Jiang**[♡], **Qun Liu**[♡]

♠Harbin Institute of Technology Shenzhen, Shenzhen, Guangdong, China
♣ Pengcheng Laboratory, Shenzhen, China
◇ Peking University, Beijing, China
♡ Huawei Noah's Ark Lab, Shenzhen, Guangdong, China

## Abstract

Training large models from scratch usually costs a substantial amount of resources. Towards this problem, recent studies such as bert2BERT and LiGO have reused small pretrained models to initialize a large model (termed the "target model"), leading to a considerable acceleration in training. Despite the successes of these previous studies, they grew pretrained models by mapping partial weights only, ignoring potential correlations across the entire model. As we show in this paper, there are inter- and intra-interactions among the weights of both the pretrained and the target models. As a result, the partial mapping may not capture the complete information and lead to inadequate growth. In this paper, we propose a method that linearly correlates each weight of the target model to all the weights of the pretrained model to further enhance acceleration ability. We utilize multi-linear operators to reduce computational and spacial complexity, enabling acceptable resource requirements. Experiments demonstrate that our method can save 76% computational costs on DeiT-base transferred from DeiT-small, which outperforms bert2BERT by +12.0% and LiGO by +20.7%, respectively.

## 1 Introduction

Transformers [47] have recently achieved great successes in various scenarios [11, 4, 20]. Generally, Transformers tend to be larger for more expressive power and better performance (e.g., ViT-G [6] and GPT-3 [3]). As the size of models continues to grow, training Transformers takes longer and can result in higher $CO_2$ emissions, which conflicts with the principles of Green AI [43]. Thus, training Transformers efficiently is crucial not only

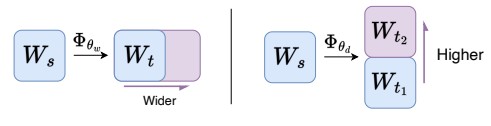

Figure 1: Expanding operators. $W_s$ and $W_t$ mean small pretrained and target weights, respectively. $\Phi_{\theta_w}$ and $\Phi_{\theta_d}$ denote width and depth expanding operators with parameters $\theta_w$ and $\theta_d$.

for financial gain but also for environmental sustainability [5]. To achieve efficient training, it is a wise option to grow a pretrained small model into a larger one, since the pretrained small model has already learned knowledge from the data, which allows for faster training compared to starting from scratch. [13]. Moreover, there are numerous pretrained models that are easily accessible [53], reducing the cost of utilizing a smaller pretrained model. Furthermore, empirical evidence shows that Transformers have inductive biases that facilitate scaling fitting [42, 22]. This demonstrates the feasibility of learning from pretrained models [53].

---

[∗]This work is done when Yu Pan is an intern at Huawei Noah's Ark Lab.
[†]Corresponding author.

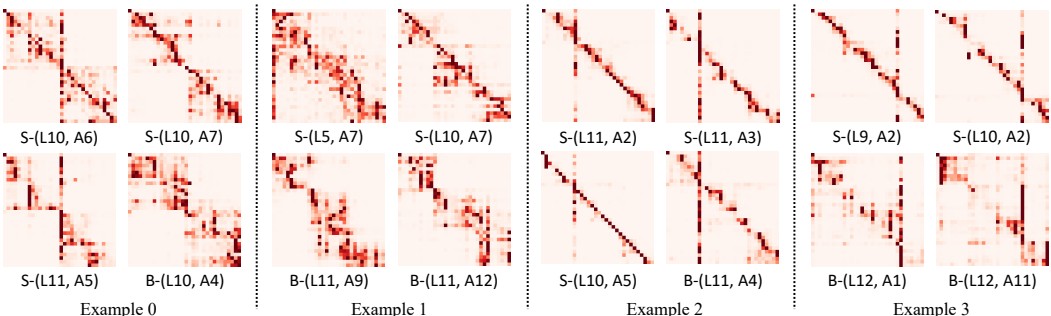

Figure 2: Attention pattern maps between BERT-Small (shorted as "S") and BERT-Base (shorted as "B") in four examples. "L" represents the layer index, while "A" denotes the index of the attention head. For example, "S-(L10, A6)" denotes the attention map from layer 10, head 6 of the BERT-Small model. The examples are derived from different sentences. In each example, we can observe the similarities of attention maps in both inter-layer and intra-layer connections, even across different scaled models.

The concept of reusing pretrained models essentially involves exploring the efficient mapping from the pretrained model to the target model. This mapping process can be represented as linear growth operators, as depicted in Figure 1. One viable approach to mapping is to leverage knowledge from other weights. For instance, StackBERT [13] utilizes low-layer information to construct new higher layers, by duplicating layers to increase depth during training epochs. bert2BERT [5] expands width through the expanding operator of Net2Net [7]. Differently, bert2BERT utilizes weights of neighbor layers for enhancing the ability, which also shows the benefit of taking knowledge from other weights. Moreover, distinct from the aforementioned research directly employing a fixed operator, LiGO [53] trains expanding operators which are tied with layers to achieve superior transfer accuracy by considering knowledge from other layers, which is advantageous for training efficiency. However, they neglect the potential connectivity between models.

**Potential Connectivity between Models.** The attention maps of BERT, as depicted in Figure 2, indicate that there are similarities not only in the weights of the same or different layers, but also across different models. These similarities suggest that we can leverage a full mapping transformation to capture this correlation in both inter-layer and intra-layer connections, even across different scaled models. By doing so, we can reuse the parameters of pretrained small models and accelerate the training of larger models. However, previous studies have overlooked the overall network connectivity and have instead expanded each weight through partial mapping transformation. For instance, Net2Net [7] only considers preserving functionality by transforming weights individually, while bert2BERT [5] increases model width head by head in Transformers. On the other hand, LiGO [53] primarily focuses on extending the weights of the same type (e.g., query, key, and value in Transformers), which heavily affects the performance of the transferred models.

**Multi-linear Operators.** Based on the above observations, we propose mapping the entire pretrained model to each weight of the target model, instead of employing a partial transformation. This approach can ensure high expressive power, promoting the potential connectivity in models for better training efficiency. However, the completed mapping has a huge parameter tensor, occupying an enormous space (see Section 3.2). To address this issue, we propose Mango, a multi-linear structure (i.e., tensor ring [33, 52, 51]), which decomposes the large mapping tensor into four smaller tensors that are bonded by ranks to construct multi-linear operators. Mango allows for efficient training by reducing the space required for the mapping tensor. Formally, Mango can be considered as a generalization of LiGO and bert2BERT (see Section 3.3). By using the full mapping approach, Mango can save 76% computational costs on DeiT-base transferred from DeiT-small, which outperforms bert2BERT [5] by +12.0% and LiGO [53] by +20.7%, respectively.

## 2 Related Work

**Efficient Training from Scratches.** Model scratches can be roughly regarded as models without knowledge priors. Some training strategies for scratches are universal and orthogonal to our method.

For example, Adam [23] and large-batch size training [66] accelerate the training process from angles of optimizer and magnitude of input data, respectively. Wang et al. [50] enable the training of very deep neural networks with limited computational resources via a technique called active memory. Shoeybi et al. [45] use mixed precision training to assist training. Low-rank methods benefit training for less memory and time [21]. Wu et al. [59] take notes of rare words for better data efficiency. Dropping layers [68], knowledge inheritance [38], and merging tokens [2] are also efficient methods for training. Another line of work [13, 63, 14, 44] is termed progressive training, which gradually increases the model size within the training process for training efficiency. Li et al. [27] employ a neural architecture search (NAS) method to search optimal sub-networks for progressive training on ViTs. Xia et al. [60] suggest a three-phase progressive training regime to achieve a good trade-off between training budget and performance. Wang et al. [55] introduces a novel curriculum learning approach for training efficiency through firstly learning simpler patterns, then progressively introducing more complex patterns.

**Efficient Training from Pretrained Models.** Pretrained models usually contain abundant data knowledge, which is helpful for training [32]. By preserving the function of a pretrained model while expanding the model size, it is feasible to give the corresponding larger model an initial state with high performance. Net2Net [7] is the first work to propose the concept of function-preserving transformations by expanding width by splitting neurons and growing depth with identity layers. However, Net2Net splits neurons randomly. Towards this problem, a series of studies [56, 58, 49, 57] propose to select the optimal subset of neurons to be split by utilizing functional steepest descent. bert2BERT [5] expands small transformers by following the function preserving idea. Recently, LiGO [53] utilizes a trainable linear operator to learn a good expanding formula. Different from these prior studies, our method tries to implement a full mapping that achieves comprehensive utilization of the whole smaller model.

**Neural Network Initialization.** Our method is related to neural network initialization techniques. Xavier [12] and Kaiming [16] initialization aim to control input variance equal to that of output. Generalizing Xavier and Kaiming methods, a universal weight initialization paradigm proposed by Pan et al. [34] can be widely applicable to arbitrary Tensorial CNNs. In addition, Hardt and Ma [15] has shown theoretically that network training benefits from maintaining identity, particularly for improving the efficiency of residual networks. Fixup [67] and ZerO [70] both set residual stem to 0 (not residual connections) to ensure the identity of signals, thereby successfully initializing ResNets. SkipInit [8] replaces Batch Normalization with a multiplier whose value is 0. ReZero [1], on the other hand, adds extra parameters of value 0 to maintain identity, resulting in faster convergence. IDInit is an initialization approach to keep the identity matrix for stable training of networks [36]. In comparison, our work explores fully reusing smaller pretrained models as efficient initialization.

**Low-rank Techniques in Neural Networks.** Low-rank methods are feasible for reducing spatial and temporal complexities in neural networks [52, 35, 62]. For example, Idelbayev and Carreira-Perpiñán [19] uses matrix decomposition to compress convolutional neural networks (CNNs) for faster inference. LoRA [17] applies low-rank matrices for fine-tuning large language models (LLMs) in affordable resources. As a parameter-efficient tuning (PETuning) method [69] for federated learning, FedPara [18] utilizes the Hadamard product on low-rank parameters for reducing communication time in federated learning. Pan et al. [33] and Li et al. [28] use tensor ring decomposition for reducing the size of neural networks. Tucker decomposition and block-term Tucker decomposition have been used in T-Net and BT-layers [24, 64] for improving the model performance, respectively. Ma et al. [30] apply block-term Tucker to compress Transformers. Yin et al. [65] propose to use ADMM to optimize tensor training to achieve better compression and performance. These techniques have built a solid foundation for our work to implement multi-linear transformation for transferring knowledge from pretrained models to target models.

## 3 Mango Operator

This section presents the proposed Mango operator. First, we provide an overview of the tensor diagram, the concept of the tensor ring matrix product operator (TR-MPO), and the formulation of Transformer architecture in Sec. 3.1. Next, we delve into the details of the proposed multi-linear mapping operator (i.e., Mango) in Sec. 3.2. Finally, we compare Mango with recent advances in terms of tensor diagrams in Sec. 3.3.

### 3.1 Preliminary

**Tensor Diagram.** Tensor decomposition is a technique that involves splitting a tensor into multiple smaller tensors, typically more than the two matrices that result from matrix factorization. To better illustrate the interactions among multiple tensors, tensor diagrams are often used. A tensor diagram is composed of two primary elements: a tensor vertex and a tensor contraction. A tensor vertex represents a tensor, and its order is determined by the number of edges connected to it. Each edge is assigned an integer that indicates the dimension of the corresponding mode. As shown in Figure 3, a 3rd-order tensor $\mathcal{T} \in \mathbb{R}^{\mathbf{i}_0 \times \mathbf{i}_1 \times \mathbf{i}_2}$ can be drawn as a circle with three edges. The process of taking the inner product of two tensors on their matching modes is called tensor contraction. For example, Two tensors, $\mathcal{M} \in \mathbb{R}^{\mathbf{i}_0 \times \mathbf{i}_1 \times \mathbf{i}_2}$ and $\mathcal{N} \in \mathbb{R}^{\mathbf{j}_0 \times \mathbf{j}_1 \times \mathbf{j}_2}$, can be contracted in corresponding positions to form a new tensor of $\mathbb{R}^{\mathbf{i}_0 \times \mathbf{i}_1 \times \mathbf{j}_2 \times \mathbf{j}_3}$, when they have equal dimensions: $\mathbf{i}_2 = \mathbf{j}_0 \triangleq e_0$. The contraction operation can be formulated as

Figure 3: Tensor diagram instances.

$$(\mathcal{M} \times_2^0 \mathcal{N})_{i_0,i_1,j_2,j_3} = \sum_{m=0}^{e_0-1} \mathcal{M}_{i_0,i_1,m} \mathcal{N}_{m,j_2,j_3}. \tag{1}$$

The tensor diagram is an elegant tool for succinctly visualizing and manipulating multi-dimensional arrays or tensors. By representing tensor operations as networks of nodes and edges, these diagrams provide an intuitive way to understand the underlying mathematical operations and their interactions. As a result, we have chosen to use tensor diagrams to illustrate our method and to analyze the connections between our approach and prior studies.

**Tensor Ring Matrix Product Operator (TR-MPO).** TR-MPO means tensor ring (TR) of an MPO [37] format. Given a $2N$-order tensor $\mathcal{X} \in \mathbb{R}^{I_1 \times J_1 \times I_2 \times J_2 ... I_N \times J_N}$, its TR-MPO decomposition can be mathematically expressed as

$$\mathcal{X}_{i_1,j_1,i_2,j_2...,i_N,j_N} \approx \sum_{r_1,r_2,...,r_N=1}^{R_1,R_2,...,R_N} \mathcal{G}^{(1)}_{r_1,i_1,j_1,r_2} \mathcal{G}^{(2)}_{r_2,i_2,j_2,r_3} \mathcal{G}^{(3)}_{r_3,i_3,j_3,r_4} \cdots \mathcal{G}^{(N)}_{r_N,i_N,j_N,r_1}, \tag{2}$$

where $\{R_1, R_2, \ldots, R_N\}$ denote the ranks, $\mathcal{G}^{(n)} \in \mathbb{R}^{R_n \times I_n \times I_n \times R_{n+1}}$ denotes a 4th-order core tensor and $R_1 = R_{N+1}$, which indicates ring-like structure.

**Transformer Architecture.** The Transformer [47] is a deep learning architecture that has revolutionized the artificial intelligence field including computer vision (CV) and natural language processing (NLP). As shown in Figure 4, a Transformer block consists of two main sub-layers: the multi-head self-attention (MHSA) layer and the feed-forward neural network (FFN) layer.

(1) MHSA Layer. The MHSA layer in the Transformer block computes the attention scores between each input element and every other element, allowing the model to attend to different parts of the input sequence during processing. Inputs of MHSA are query matrix $\mathbf{Q} \in \mathbb{R}^I$, key matrix $\mathbf{K} \in \mathbb{R}^I$ and value matrix $\mathbf{V} \in \mathbb{R}^I$ with parameters $\mathbf{W}^Q \in \mathbb{R}^{I \times O}, \mathbf{W}^K \in \mathbb{R}^{I \times O}, \mathbf{W}^V \in \mathbb{R}^{I \times O}, \mathbf{W}^O \in \mathbb{R}^{I \times O}$. Usually, it constrains $I = O$. Moreover, MHSA separates these parameters into $n$ heads: $\{\mathbf{W}^{Q,i}\}^n, \{\mathbf{W}^{K,i}\}^n, \{\mathbf{W}^{V,i}\}^n, \{\mathbf{W}^{O,i}\}^n$. The MHSA mechanism can be formulated as follows:

$$\text{Att}_i(\mathbf{Q}, \mathbf{K}, \mathbf{V}) = \text{softmax}\left(\frac{\mathbf{Q}\mathbf{W}^{Q,i}(\mathbf{K}\mathbf{W}^{K,i})^T}{\sqrt{d_k}}\right)\mathbf{V}\mathbf{W}^{V,i}\mathbf{W}^{O,i^T},$$

$$\text{MHSA}((\mathbf{Q}, \mathbf{K}, \mathbf{V})) = \sum_{i=1}^{n} \text{Att}_i(\mathbf{Q}, \mathbf{K}, \mathbf{V}), \tag{3}$$

where $d_k$ is the dimensionality of the key vectors. The self-attention mechanism is performed multiple times in parallel, each time using a different set of learned parameters $\mathbf{W}^{Q,i}$, $\mathbf{W}^{K,i}$, and $\mathbf{W}^{V,i}$ to compute multiple "heads" of attention. The resulting attention heads are linearly transformed by a learned weight matrix $\mathbf{W}^{O,i}$ to produce the output. At last, MHSA concatenates the output as the final result of the MHSA layer.

(2) FFN Layer. The FFN layer in the Transformer block is responsible for applying a non-linear transformation to the output of the self-attention layer. $\mathbf{X} \in \mathbb{R}^I$ is the input. The weights are

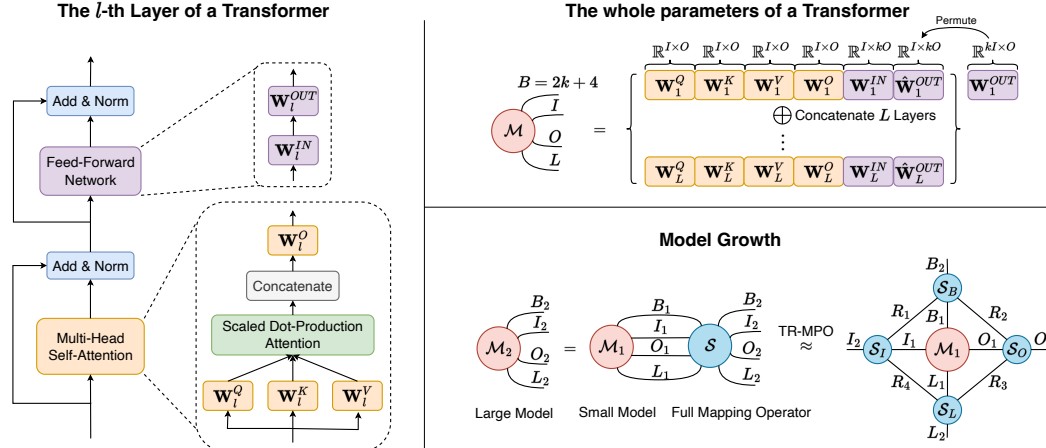

Figure 4: The full mapping of the Mango operator. The left sub-figure shows that the parameters of a Transformer layer are $\mathbf{W}^Q, \mathbf{W}^K, \mathbf{W}^V, \mathbf{W}^O, \mathbf{W}^{IN}$ and $\mathbf{W}^{OUT}$. $I$ means an input dimension size. $O$ denotes an output dimension size. $L$ is the layer number. We concatenate all the parameters into a tensor $\mathcal{M}$ and then consider a full mapping operator $\mathcal{S}$ to transform this tensor. However, $\mathcal{S}$ is huge, thereby, we use a multi-linear method TR-MPO to decompose it into four smaller tensors $\{\mathcal{S}_B, \mathcal{S}_I, \mathcal{S}_O, \mathcal{S}_L\}$ form the Mango operator $\Phi_{\mathcal{S}_B, \mathcal{S}_I, \mathcal{S}_O, \mathcal{S}_L}$.

$\mathbf{W}^{IN} \in \mathbb{R}^{I \times kO}$ and $\mathbf{W}^{OUT} \in \mathbb{R}^{kI \times O}$, where usually $I = O$ and $k$ is a ratio that is often set to 4. The FFN layer can be formulated as

$$\text{FFN}(\mathbf{X}) = \text{GeLU}(\mathbf{X}\mathbf{W}^{IN})\mathbf{W}^{OUT}. \tag{4}$$

We neglect biases in the formulation as it is usually set to 0 at initialization. The output of the FFN layer is obtained by applying two linear transformations to the output of the MHSA layer.

Finally, both the self-attention output and the FFN output are processed through a residual connection and layer normalization to prevent the model from collapsing or overfitting to the training data. Apparently, the parameters in a Transformer are mainly based on its linear transformation matrices, i.e., $\mathbf{W}^Q, \mathbf{W}^K, \mathbf{W}^V, \mathbf{W}^O, \mathbf{W}^{IN}$ and $\mathbf{W}^{OUT}$.

## 3.2 Multi-linear Operator

The essential of model growth is to transfer the knowledge from the small model to the bigger counterpart. Prior study [5] finds that taking weights from the neighbor layer as initial parameters can further improve the convergence speed, which gives the insights that knowledge from other layers can help training as there are many similar attention maps among layer. Nevertheless, the fact is that this similarity exists all over the pretrained model as shown in Figure 2, which motivates us to consider the possibility of whether we can utilize the knowledge from all weights. Based on this consideration, we construct a mapping of the whole model. As the correlation is hard to formulate heuristically, we learn the mapping parameter to implicitly transfer the pretrained model.

**Notation.** Here, we give the notations for clearly formulating our method. A model with $L$ layers and a $D$ hidden size can be denoted as $\mathbf{M}(L, D)$. Following Section 3.1, parameters of $j$-th layer are a set $\boldsymbol{\theta}_j = \{\mathbf{W}_j^Q \in \mathbb{R}^{I \times O}, \mathbf{W}_j^K \in \mathbb{R}^{I \times O}, \mathbf{W}_j^V \in \mathbb{R}^{I \times O}, \mathbf{W}_j^O \in \mathbb{R}^{I \times O}, \mathbf{W}_j^{IN} \in \mathbb{R}^{I \times kO}, \mathbf{W}_j^{OUT} \in \mathbb{R}^{kI \times O}\}$, $j \in [L]$, usually satisfying $I = O = D$. The weight of $\mathbf{M}(L, D)$ is $\boldsymbol{\theta}^{L,D} = \{\theta_j\}_{j=1}^L$. A growth operator with parameter $\mathcal{S}$ is denoted as $\Phi_{\mathcal{S}}$. A mapping from $\mathbf{M}(L_1, D_1) \rightarrow \mathbf{M}(L_2, D_2)$ can be denoted as $\boldsymbol{\theta}^{L_2,D_2} = \Phi_{\mathcal{S}}(\boldsymbol{\theta}^{L_1,D_1})$. In the case of $L_1 < L_2$, this mapping represents a growth mapping. After growing, $\boldsymbol{\theta}^{L_2,D_2}$ will be used as initial weights for the target model.

**Full Mapping Operator.** To utilize all weights for knowledge transferring, we first concatenate weights across layers. As shown in Figure 4, we concatenate $j$-th layer to form a tensor of shape $B \times I \times O$ along with order $I$ and $O$ where $B = 2k + 4$. Then the final weight tensor $\mathcal{M} \in \mathbb{R}^{B \times I \times O \times L}$ can be derived by combining the concatenated tensors of $L$ layers. Giving

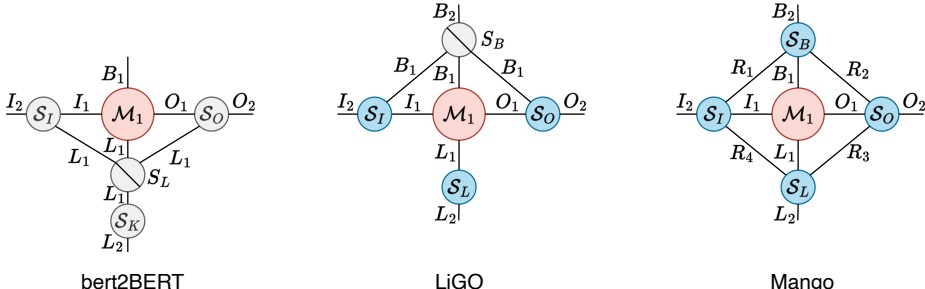

Figure 5: Growth processes with the tensor diagram. The circle with an oblique stroke means a super-diagonal tensor. The blue color means a trainable operator, while the gray color denotes an untrainable operator. The red color means the smaller model. Each tensor diagram means the growth process that growing $M_1$ to a bigger model through growth the operator $S_*$.

a small model $\boldsymbol{\mathcal{M}}_1 \in \mathbb{R}^{B_2 \times I_1 \times O_1 \times L_1}$, a big model $\boldsymbol{\mathcal{M}}_2 \in \mathbb{R}^{B_1 \times I_2 \times O_2 \times L_2}$, and a parameter $\boldsymbol{\mathcal{S}} \in \mathbb{R}^{B_1 \times I_1 \times O_1 \times L_1 \times B_2 \times I_2 \times O_2 \times L_2}$ of a full mapping operator $\Phi_{\boldsymbol{\mathcal{S}}}$, we can transfer $\boldsymbol{\mathcal{M}}_1$ to $\boldsymbol{\mathcal{M}}_2$ as

$$(\boldsymbol{\mathcal{M}}_2)_{b_2,i_2,o_2,l_2} = \sum_{b_1,i_1,o_1,l_1} (\boldsymbol{\mathcal{M}}_1)_{b_1,i_1,o_1,l_1} \boldsymbol{\mathcal{S}}_{b_1,i_1,o_1,l_1,b_2,i_2,o_2,l_2}, \tag{5}$$

where $b_1$, $i_1$, $o_1$, $l_1$, $b_2$, $i_2$, $o_2$, and $l_2$ are entries of corresponding tensors.

**Multi-linear Mapping Operator.** Note that $\boldsymbol{\mathcal{S}}$ is extremely big, which makes it infeasible to display this mapping by applying a whole transformation tensor. Therefore, we propose a multi-linear operator named Mango which decomposes $\boldsymbol{\mathcal{S}}$ through a tensor ring matrix product operator (TR-MPO) in four small tensors $\{\boldsymbol{\mathcal{S}}_B \in \mathbb{R}^{R_1 \times B_1 \times B_2 \times R_2}, \boldsymbol{\mathcal{S}}_O \in \mathbb{R}^{R_2 \times O_1 \times O_2 \times R_3}, \boldsymbol{\mathcal{S}}_L \in \mathbb{R}^{R_3 \times L_1 \times L_2 \times R_4}, \boldsymbol{\mathcal{S}}_I \in \mathbb{R}^{R_4 \times I_1 \times I_2 \times R_1}\}$. $R = \{R_1, R_2, R_3, R_4\}$ is the rank of TR-MPO. Then, we can update $\Phi_{\boldsymbol{\mathcal{S}}}$ to $\Phi_{\boldsymbol{\mathcal{S}}_B, \boldsymbol{\mathcal{S}}_I, \boldsymbol{\mathcal{S}}_O, \boldsymbol{\mathcal{S}}_L}$, and Eq. (5) can be reformulated with a multi-linear form as

$$(\boldsymbol{\mathcal{M}}_2)_{b_2,i_2,o_2,l_2} = \sum_{b_1,i_1,o_1,l_1,r_1,r_2,r_3,r_4} (\boldsymbol{\mathcal{M}}_1)_{b_1,i_1,o_1,l_1} (\boldsymbol{\mathcal{S}}_B)_{r_1,b_1,b_2,r_2} (\boldsymbol{\mathcal{S}}_O)_{r_2,o_1,o_2,r_3} (\boldsymbol{\mathcal{S}}_L)_{r_3,l_1,l_2,r_4} (\boldsymbol{\mathcal{S}}_I)_{r_4,i_1,i_2,r_1}. \tag{6}$$

The total size of $\boldsymbol{\mathcal{S}}_B$, $\boldsymbol{\mathcal{S}}_I$, $\boldsymbol{\mathcal{S}}_O$, and $\boldsymbol{\mathcal{S}}_L$ is exponentially less than $\boldsymbol{\mathcal{S}}$, which makes Mango viable for practical implementation while maintaining the full correlation between the small and big models.

**Training Target.** After designing the multi-linear operators, we train these operators to obtain the function preserving $\boldsymbol{\mathcal{M}}_2$. The training target can be denoted as

$$\min_{\boldsymbol{\mathcal{S}}_B, \boldsymbol{\mathcal{S}}_I, \boldsymbol{\mathcal{S}}_O, \boldsymbol{\mathcal{S}}_L} \mathbb{E}_{\mathbf{X} \sim \mathcal{D}} \mathcal{L}(\mathbf{X}, \boldsymbol{\mathcal{M}}_2), \ \ \text{w.r.t} \ \boldsymbol{\mathcal{M}}_2 = \Phi_{\boldsymbol{\mathcal{S}}_B, \boldsymbol{\mathcal{S}}_I, \boldsymbol{\mathcal{S}}_O, \boldsymbol{\mathcal{S}}_L}(\boldsymbol{\mathcal{M}}_1), \tag{7}$$

where $\mathcal{L}$ is a loss function, and $\mathcal{D}$ is a data distribution. By replacing the full space of $\boldsymbol{\mathcal{S}}$ with four small spaces, the spatial requirements of the training process are reduced exponentially.

**Procedures of Applying Mango.** The procedures of Mango can be divided into four steps: (i) concatenating weights $\theta^{L_1,D_1}$ of a pretrained model $\mathbf{M}(L_1, D_1)$ to construct a tensor $\boldsymbol{\mathcal{M}}_1$; (ii) training the growth operator $\Phi_{\boldsymbol{\mathcal{S}}_B, \boldsymbol{\mathcal{S}}_I, \boldsymbol{\mathcal{S}}_O, \boldsymbol{\mathcal{S}}_L}$ items of Eq. (7) in a few steps (e.g., 100) to make transferred models maintaining function; (iii) recovering weight tensor $\boldsymbol{\mathcal{M}}_2$ through the multi-linear operator $\Phi_{\boldsymbol{\mathcal{S}}_B, \boldsymbol{\mathcal{S}}_I, \boldsymbol{\mathcal{S}}_O, \boldsymbol{\mathcal{S}}_L}$; (iv) splitting $\boldsymbol{\mathcal{M}}_2$ to the weight $\theta^{L_2,D_2}$ of the target model $\mathbf{M}(L_2, D_2)$ as initialization to continue training.

**Remark.** Tensor decomposition is often used to compress neural networks. Actually, there are spaces for further reducing the operator size. However, we are not exploring the border of compression ratio, but the possibility of the learning ability of multi-linear operators and the influence on model growth. Therefore, we decompose the huge $\boldsymbol{\mathcal{S}}$ into four interpretable smaller tensors. $\boldsymbol{\mathcal{S}}_B$ means the interactions on the parameters in the same layer. $\boldsymbol{\mathcal{S}}_I$ and $\boldsymbol{\mathcal{S}}_O$ denote the transformations of input and output dimensions in one parameter, respectively. $\boldsymbol{\mathcal{S}}_L$ indicates the relationship among layers. $R$ means the low-rank level of $\boldsymbol{\mathcal{S}}$. Smaller $R$ means less correlation of these four tensors.

### 3.3 Comparison with Recent Advances

Basically, the goal of model growth is to widen and deepen the width and depth of the pretrained models, and operation on width and depth can be easy to implement. Therefore, most of the previous studies grow width and depth separately, although there may exist some correlation that may influence the training efficiency. In this part, we analyze the difference among methods of model growth with the help of the tensor diagrams.

We illustrate a comparison among bert2BERT, LiGO, and Mango in Figure 5. The red circle means a pretrained model, while the blue and gray circles denote trainable and untrainable operator parameters, respectively. The circle with an oblique stroke represents a super-diagonal tensor, where the diagonal elements are all 1. This super-diagonal tensor means that growth operators expand other modes along the mode of the super-diagonal tensor.

Table 1: The comparison among bert2BERT, LiGO and Mango. In a case of $\mathbf{M}(L_1, D_1) \to \mathbf{M}(L_2, D_2)$, there usually are $I_1 = O_1 = D_1$ and $I_2 = O_2 = D_2$. $R = \max(R_1, R_2, R_3, R_4)$.

| Method | Reference | Operator Parameter | Trainability | Spatial Complexity |
|--------|-----------|--------------------|--------------|--------------------|
| bert2BERT | Chen et al. [5] | $\mathcal{S}_I, \mathcal{S}_O, \mathcal{S}_K$ | ✗ | $2L_1 D_1 D_2 + L_1 L_2$ |
| LiGO | Wang et al. [53] | $\mathcal{S}_I, \mathcal{S}_O, \mathcal{S}_L$ | ✓ | $2B_1 D_1 D_2 + L_1 L_2$ |
| Mango | - | $\mathcal{S}_B, \mathcal{S}_I, \mathcal{S}_O, \mathcal{S}_L$ | ✓ | $2RD_1 D_2 + R^2(B_1 B_2 + L_1 L_2)$ |

**bert2BERT.** As shown in Figure 5, the parameters in the growth operator of bert2BERT are all frozen, since they are designed by heuristic inspiration to preserve function. bert2BERT expands modes of $I_1$ and $O_1$ along with the mode of $L_1$, indicating that one weight is expanded without knowledge from any other weight. To further increase the training ability, bert2BERT applies $\mathcal{S}_K$ to construct Advanced Knowledge Initialization (AKI) to utilize knowledge of other layers to help accelerate training.

**LiGO.** Different from bert2BERT, the LiGO operator can be trained. In addition, LiGO expands $I_1$ and $O_1$ along the mode of $B_1$. With $\mathcal{S}_L$, LiGO can combine knowledge among layers. Operators of Net2Net and StackBERT grow with or depth separately, which can be regarded as a sub-solution to $\mathcal{S}_I$, $\mathcal{S}_O$, and $\mathcal{S}_L$ of LiGO. However, one weight in LiGO will not leverage the knowledge of weights in the same layer. Therefore, LiGO only implements partial mapping like bert2BERT.

**Mango.** We employ Mango on each mode of the weight $\mathcal{M}_1$. Mango can approach the full mapping tensor $\mathcal{S}$ as a low-rank approximation with rank $R$, rather than partial mapping operators like LiGO and bert2BERT. Therefore, Mango can obtain $\mathcal{M}_2$ to capture adequate knowledge from pretrained weights as formulated in Eq. 6. Moreover, the diagonal tensor in LiGO and contraction between a diagonal tensor and $\mathcal{S}_K$ in bert2BERT are subsets of $\mathcal{S}_L$ and $\mathcal{S}_B$ in Mango. Therefore, Mango can be a generalization of bert2BERT and LiGO with more expressive power.

## 4 Experiment

In this section, we design a set of experiments to validate the proposed Mango. To begin with, we conduct an ablation study to analyze the influence of Mango on width and depth in Sec. 4.1. Then we conduct image classification with DeiT-B [46] on ImageNet to show the acceleration in computer vision task in Sec. 4.2. Later we conduct a pretraining experiment with BERT-Base [10] to demonstrate the effectiveness in natural language processing tasks in Sec. 4.3. At last, we also employ a pretraining experiment with GPT-Base [39] to show the wide adaption of Mango in Sec. 4.4.

**Ratio of saving FLOPs.** We evaluate the training efficiency in terms of the ratio of saved floating-point operations (FLOPs). Let us assume that training a model from scratch to convergence on metric $\Psi$ (e.g., MLM loss or accuracy) necessitates $\xi_{Scratch}$ FLOPs. Then, for a method that reaches the same metric $\Psi$ with $\xi_*$ FLOPs, its FLOP saving ratio $r$ can be computed as

$$r = \frac{\xi_{Scratch} - \xi_*}{\xi_{Scratch}}. \tag{8}$$

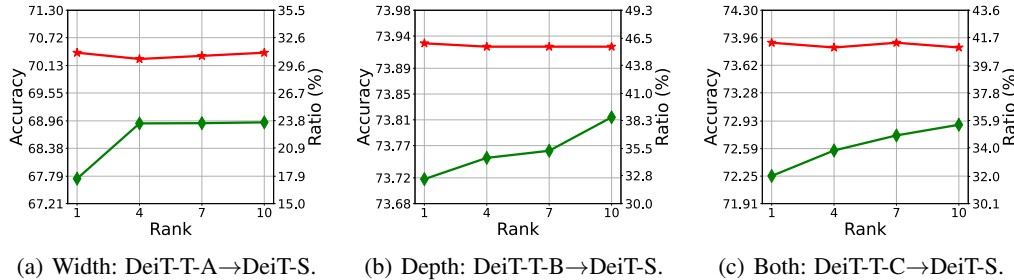

| (a) Width: DeiT-T-A→DeiT-S. | (b) Depth: DeiT-T-B→DeiT-S. | (c) Both: DeiT-T-C→DeiT-S. |

Figure 6: Influence from ranks on expanding width, depth, and both of them. The green curve means the accuracy of training operators for 100 steps. The red curve means the final acceleration ratio on training DeiT-S. Mango can achieve training acceleration in every case. Moreover, in each sub-graph, when the accuracy increases with a higher rank, the acceleration ratio keeps almost fixed.

## 4.1 Ablation Study

In this experiment, we explore the influence of Mango on growing width, depth and both of them in addition to rank setting. We use three tiny vision Transformers (ViTs) [11], i.e., DeiT-T-A, DeiT-T-B, and DeiT-T-C, for growing to DeiT-S [46] on ImageNet [9]. Structures of these DeiTs can be found in Table 4 of the Appendix. For ease of setting, we set all the ranks the same in the range of $\{1, 4, 7, 10\}$. We train Mango operators for 100 steps, which only requires negligible time. We use Adam with learning rate 1e-3 and weight decay 1e-2 for 300 epoch optimization. The batch size is 1024.

Results are shown in Figure 6. Mango achieves acceleration of at most 31.0%, 46.0%, and 41.3% on expanding width, depth, and both of them, respectively, which shows Mango can fit various growth scenarios. Interestingly, these acceleration ratios are higher when the operator accuracies are better along the types of pretrained models, e.g., as DeiT-T-B at most achieves 73.81% accuracy which is higher than 72.89% of DeiT-T-A, DeiT-T-B derives higher acceleration ratio. This phenomenon suggests that when there are multiple pretrained models can be selected, the better accuracy for one pretrained model through Mango, the faster training speed can be obtained for the target models.

In each expanding case, the accuracies of operator training tend to increase with higher ranks. However, all the cases on the three pretrained models show that better accuracy will not lead to faster training with the same pretrained model. For example, in Figure 6(a), the operator with rank 10 has an accuracy that is 1.19% higher than the operator with rank 1. Nevertheless, the two ranks reach the same acceleration ratio. This result suggests that rank 1 is enough to use, and also enjoys the advantages of spatial and temporal complexities compared to bigger ranks. And we choose rank 1 for Mango to construct the later experiments.

## 4.2 Results on Large-Scale Vision Transformer

We conduct the experiment to show the training acceleration on large-scale vision Transformers. We train the Deit-B from Deit-S [46] on ImageNet. Structures of the two models are shown in Table 4 of the Appendix. Ranks of Mango are all set to 1 for complexity benefit without performance loss. The operators of Mango and Ligo are all trained within 100 steps. We use Adam as the optimizer with learning rate 1e-3 and weight decay 1e-2. The batch size is 1024. The training epoch is 300.

Results are shown in Figure 7(a). Mango saves 76% FLOPs from Scratch which converges to an accuracy of 80.45% in the end. Compared with schemes of training from Scratch (i.e., Srcatch and StackBERT), methods of training from a smaller model (i.e., Mango, bert2BERT, and LiGO) can attain 70% accuracy in a short time, even bert2BERT start in low accuracy. Compared with the recent SOTA models, Mango has surpassed bert2BERT for +12.0%, and LiGO for +20.7%. This experiment shows the ability of Mango to achieve significant improvement in training acceleration. To investigate the influence of Mango on transferring ability, we also conduct an experiment on downstream tasks, including CIFAR10 [26], CIFAR100 [26], Flowers [31], Cars [25], and ChestXRay8 [54]. Results are shown in Table 2. It is easy to see that Mango achieves similar results to the Scratch, which indicates Mango has not influenced the transferring ability.

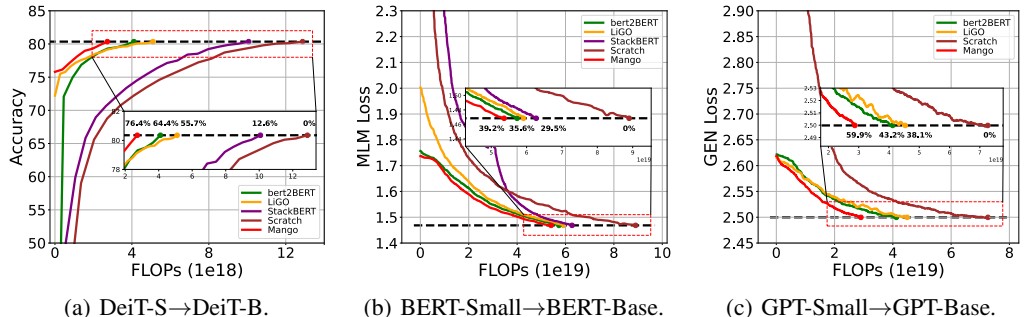

| (a) DeiT-S→DeiT-B. | (b) BERT-Small→BERT-Base. | (c) GPT-Small→GPT-Base. |

Figure 7: Results of pretraining DeiT-B, BRET-Base and GPT-Base. Compared to baselines, Mango can achieve the highest savings in FLOPs with 76.4% for DeiT-B, 39.2% for BERT-Base, and 59.9% for GPT-Base from the Scratch model. We also illustrate the corresponding results on wall time in Figure 10 of the appendix.

Table 2: Results of transfer learning performance of DeiT-B. Mango can achieve similar performance to the Scratch model in downstream tasks while saving 76.4% FLOPs.

| Method | FLOPs ($\times$ 1e18) | Ratio (Saving) | CIFAR10 | CIFAR100 | Flowers | Cars | ChestXRay8 | Average |
|---|---|---|---|---|---|---|---|---|
| *Training from Scratch* | | | | | | | | |
| Scratch | 12.9 | - | 99.03 | 90.22 | 97.27 | 91.89 | 55.66 | 86.82 |
| StackBERT | 11.3 | 12.6% | 99.11 | 90.10 | 97.44 | 91.71 | 55.63 | 86.80 |
| *Training from the Pretrained Model:* $\mathbf{M}(12, 384) \rightarrow \mathbf{M}(12, 768)$ | | | | | | | | |
| bert2BERT | 4.6 | 64.4% | 98.99 | 90.47 | 97.51 | 91.88 | 55.34 | 86.84 |
| LiGO | 5.7 | 55.7% | 99.11 | 90.52 | 97.18 | 91.82 | 55.45 | 86.82 |
| Mango | 3.0 | **76.4%** | 99.13 | 90.23 | 97.49 | 91.83 | 55.46 | 86.83 |

## 4.3 Pretraining on BERT

In this experiment, we conduct the validation to show the training acceleration on BERT [10, 61]. The dataset is the concatenation of English Wikipedia and Toronto Book Corpus [71]. We train the BERT-Base from BERT-Small. The training epoch is 40. The batch size is 768. We list the structures of the two models in Table 5 of the Appendix. The ranks of Mango are all 1. Mango and LiGO are both warmly trained for 100 steps. The optimizer is set to AdamW. The learning rate is 1e-4 and the weight decay is 1e-2.

We illustrate the training curves in Figure 7(b). The methods of training from BERT-Small (i.e., Mango, bert2BERT, and LiGO) can all surpass the progressive training StackBERT of acceleration ratio 29.5%. bert2BERT is over StackBERT by +6.1%. Mango achieves the highest acceleration of 39.2% FLOPs which is +3.6% more than bert2BERT. Loss of StackBERT steeply decreases within training for Stacking layers, which can also demonstrate the pretrained weights are helpful for training efficiency. We show the effectiveness of Mango on SQuAD and GLUE benchmark as in Table 3. Mango shows the same transferring ability with faster convergence, which indicates a promising use for practical training.

## 4.4 Pretraining on GPT

We also implement the experiment to show the training acceleration on GPT [39]. The dataset is the concatenation of English Wikipedia and Toronto Book Corpus [71]. We train the GPT-Base from GPT-Small. The structures of the two models are shown in Table 5 of the Appendix. The ranks of Mango are all 1. Mango and LiGO are trained for 100 steps. We use Adamw with learning rate 1e-4 and weight decay 1e-2. The batch size is 512. The training epoch is 35.

Table 3: Results of downstream tasks of BERT-Base on GLUE [48], SQuADv1.1 [40], and SQuADv2.0 [41] dataset. Mango can also achieve similar performance with the Scratch model while enjoying training efficiency.

| Model | FLOPs (× 1e19) | Ratio (Saving) | SQuADv1.1 (F1) | SQuADv2.0 (F1) | SST-2 (Acc) | MNLI (Acc) | MRPC (Acc) | COLA (Mcc) | QNLI (Acc) | STS-B (Acc) | QQP (Acc) | GLUE Avg. | SQuAD Avg. |
|---|---|---|---|---|---|---|---|---|---|---|---|---|---|
| *Training from Scratch* | | | | | | | | | | | | | |
| Scratch | 8.9 | - | 89.21 | 77.90 | 92.18 | 84.19 | 87.55 | 56.35 | 91.50 | 89.16 | 90.25 | 84.45 | 83.56 |
| StackBERT | 6.3 | 29.5% | 89.82 | 78.21 | 92.94 | 84.63 | 87.65 | 61.61 | 90.95 | 87.13 | 90.20 | 85.01 | 84.01 |
| *Training from the Pretrained Model:* $\mathbf{M}(12, 384) \rightarrow \mathbf{M}(12, 768)$ | | | | | | | | | | | | | |
| bert2BERT | 5.7 | 35.6% | 90.02 | 78.99 | 92.89 | 84.92 | 86.91 | 60.32 | 91.81 | 88.11 | 90.72 | 85.10 | 84.50 |
| LiGO | 5.9 | 33.5% | 90.09 | 78.34 | 92.75 | 84.99 | 87.44 | 61.10 | 91.33 | 87.94 | 90.42 | 85.14 | 84.22 |
| Mango | 5.4 | **39.2%** | 90.17 | 78.77 | 92.71 | 84.86 | 87.94 | 62.88 | 91.49 | 88.73 | 90.62 | 85.60 | 84.47 |

We compare the Scratch model, bert2BERT, LiGO, and Mango in Figure 7(c). We observe that the proposed Mango achieves a 59.9% acceleration ratio. While GPT is different from BERT in different structures, including layer normalization, mask method, and training strategy, Mango can always keep the highest performance. Although LiGO is lower than bert2BERT at the beginning, it converges slower and reaches 38.1% at last. By contrast, the Mango is almost at the lowest loss in the whole training process and achieves +16.7% more than bert2BERT and +21.8% more than LiGO, which shows a significant acceleration than the baselines.

## 5   Conclusion

Training Transformers can pose a significant demand on computational resources. Reusing pretrained models as an initialization strategy for the target model offers a promising approach to reducing resource costs and accelerating training. However, previous studies only mapped partial weights when growing models, which may fail to consider potential correlations in the whole model and result in inadequate growth mapping. Inspired by this observation, we propose to consider the interaction among all weights in the model to further improve acceleration ability. Specifically, we utilize a full mapping to comprehensively reuse the pretrained model, taking into account all correlations between model weights. As the full mapping tensor is huge and cannot be employed in practice, we propose to use Mango, a multi-linear operator, to reduce computation and space complexity. Experimental results demonstrate that Mango consistently achieves significant acceleration on various large-scale models (e.g., DeiT-B, BERT-Base, and GPT-Base). In the future, we hope that this method can contribute to green AI and significantly reduce the cost of training Transformers.

**Limitation.** While Mango significantly reduces training costs for large models, it still requires weights of a small pretrained model as the prior knowledge. Additionally, the Mango operator necessitates extra training for initializing. However, obtaining small model weights is relatively simple within the community, and given the substantial cost savings, the resources required for training the operator are minimal.

**Societal Impact.** The acceleration from Mango significantly impacts energy conservation and environmental protection, promoting the growth of Green AI, which boosts efficiency and reduces computational demand, minimizing energy usage and carbon emissions. Mango can also democratize AI, allowing broader access to technologies of pertaining models without the need for large resources, and promoting sustainable and accessible AI solutions that respect our environmental limits.

## Acknowledgements

This work was partially supported by the National Key Research and Development Program of China (No. 2018AAA0100204), a key program of fundamental research from Shenzhen Science and Technology Innovation Commission (No. JCYJ20200109113403826), the Major Key Project of PCL (No. 2022ZD0115301), and an Open Research Project of Zhejiang Lab (NO.2022RC0AB04).

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
