# Appendix

## A Additional Experiments and Details

### A.1 Experiment Structures

We show DeiT structures in Table 4, and structures of BERT and GPT in Table 5.

Table 4: The structures of DeiT.

| Config | DeiT-T-A | DeiT-T-B | DeiT-T-C | DeiT-S | DeiT-B |
|---|---|---|---|---|---|
| # layers | 12 | 8 | 10 | 12 | 12 |
| # hidden | 192 | 384 | 320 | 384 | 768 |
| # heads | 3 | 6 | 5 | 6 | 12 |
| input size | 224 | 224 | 224 | 224 | 224 |
| patch size | 16 | 224 | 16 | 16 | 16 |

Table 5: The structures of BERT and GPT.

| Config | BERT-Small | BERT-Base | BERT-Large | GPT-Small | GPT-Base |
|---|---|---|---|---|---|
| # layers | 12 | 12 | 24 | 12 | 12 |
| # hidden | 512 | 768 | 1024 | 512 | 768 |
| # heads | 8 | 12 | 16 | 8 | 12 |
| # vocab | 30522 | 30522 | 30522 | 50257 | 50257 |
| seq. length | 512 | 512 | 512 | 1024 | 1024 |

### A.2 Additional Experiment of Growing Swin-T to Swin-S

We also conduct an experiment to show the training acceleration on Swin-Transformers [29][1]. We train Swin-S from Swin-T with 128 batch size for 240 epochs. The training dataset is ImageNet. Ranks of Mango are all set to 1. The operators of Mango and Ligo are all trained within 100 steps. We use Adamw as the optimizer with a learning rate 1e-3 and weight decay 1e-8. The learning rate is reduced through a cosine scheme.

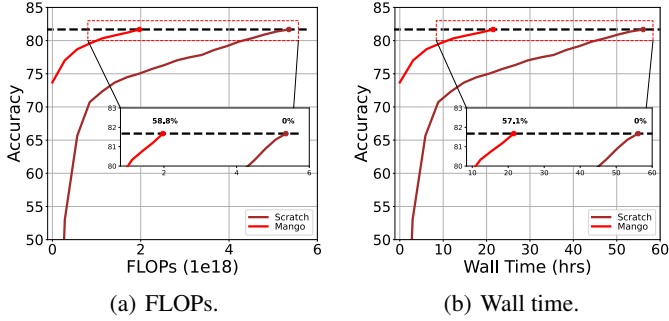

(a) FLOPs.  (b) Wall time.

Figure 8: Results on Swin-T→Swin-S.

Results are shown in Figure 8. Mango saves 58.8% FLOPs and 57.1% wall time from Scratch which converges to an accuracy of 81.67%. Showing the effectiveness of Mango on Swin-Transformers, we demonstrate the potential of Mango to serve as a general-purpose growth operator.

---

[1]This experiment is based on the code at: https://github.com/microsoft/Swin-Transformer.

### A.3 Additional Experiment of Growing BERT-Base to BERT-Large

We also conduct an experiment to show the training acceleration on BERT-Large form BERT-Base with 512 batch size for 3 epochs. The structures of the two models are shown in Table 5 of the Appendix. Other settings follow Sec. 4.3.

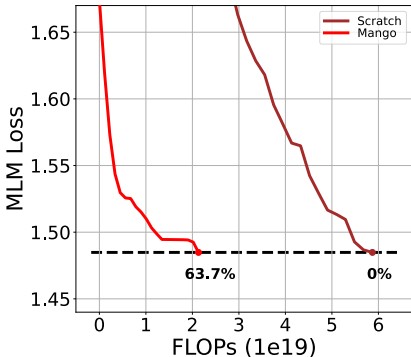

Figure 9: Results on BERT-Base→BERT-Large.

We illustrate the training curves in Figure 9. Mango achieves 63.7% acceleration in an early training stage, indicating consistent acceleration to other experiments. This experiment shows that the proposed Mango can perform the same training efficiency even in a huge Transformer model.

### A.4 Details of Training Mango

In experiments of DeiTs, we train Mango operators with Adam optimizer for 100 steps on ImageNet. The learning rate is 1e-4. Minibatch is 1536.

In experiments of BERT and GPT models, we train Mango operators with Adamw optimizer for 100 steps on the concatenation of English Wikipedia and Toronto Book Corpus. The learning rate is 1e-5. Minibatch is 512.

### A.5 Details of Downstream Tasks

In downstream tasks of DeiTs, we use Adam with a learning rate chosen from {8e-5, 1e-4}. The batch size is 256, and we run up to 100 epochs. We run each experiment three times for calculating mean and standard derivation. Detailed results are shown in Table 6.

In downstream tasks of BERT, we set the batch size to 32 and use Adam with the learning rate from {5e-6, 1e-5, 2e-5, 3e-5} and epochs from {4, 5, 10} for the GLUE tasks fine-tuning. For the SQuAD fine-tuning, we set the batch size to 16 and the learning rate to 3e-5, and train for 4 epochs. All results are the average of 5 runs on the dev set. Detailed results are shown in Table 7 and Table 8.

### A.6 Wall Time of Experiments

We illustrate the acceleration of training DeiT-B, BERT-Base and GPT-Base in terms of wall time in Figure 10, which corresponds to Figure 7.

Table 6: Detailed downstream results on the ImageNet.

| Method | FLOPs (× 1e18) | Ratio (Saving) | CIFAR10 | CIFAR100 | Flowers | Cars | ChestXRay8 | Average |
|---|---|---|---|---|---|---|---|---|
| *Training from Scratch* | | | | | | | | |
| Scratch | 12.9 | - | 99.03 (0.08) | 90.22 (0.27) | 97.27 (0.17) | 91.89 (0.53) | 55.66 (0.23) | 86.82 (0.26) |
| StackBERT | 11.3 | 12.6% | 99.11 (0.11) | 90.10 (0.28) | 97.44 (0.25) | 91.71 (0.28) | 55.63 (0.38) | 86.80 (0.26) |
| *Training from the Pretrained Model:* $\mathbf{M}(12, 384) \rightarrow \mathbf{M}(12, 768)$ | | | | | | | | |
| bert2BERT | 4.6 | 64.4% | 98.99 (0.04) | 90.47 (0.19) | 97.51 (0.07) | 91.88 (0.47) | 55.34 (0.22) | 86.84 (0.20) |
| LiGO | 5.7 | 55.7% | 99.11 (0.08) | 90.52 (0.33) | 97.18 (0.27) | 91.82 (0.36) | 55.45 (0.26) | 86.82 (0.26) |
| Mango | 3.0 | **76.4%** | 99.13 (0.06) | 90.23 (0.24) | 97.49 (0.15) | 91.83 (0.34) | 55.46 (0.36) | 86.83 (0.23) |

Table 7: Detailed downstream results on the GLUE benchmark.

| Model | FLOPs (× 1e19) | Ratio (Saving) | SST-2 (Acc) | MNLI (Acc) | MRPC (Acc) | COLA (Mcc) | QNLI (Acc) | STS-B (Acc) | QQP (Acc) | GLUE Avg. |
|---|---|---|---|---|---|---|---|---|---|---|
| *Training from Scratch* | | | | | | | | | | |
| Scratch | 8.9 | - | 92.18(0.09) | 84.19(0.17) | 87.55(0.29) | 56.35(1.93) | 91.50(0.09) | 89.16(0.28) | 90.25(0.13) | 84.45(0.42) |
| StackBERT | 6.3 | 29.5% | 92.94(0.06) | 84.63(0.19) | 87.65(0.20) | 61.61(3.58) | 90.95(0.10) | 87.13(0.60) | 90.20(0.15) | 85.01(0.70) |
| *Training from the Pretrained Model:* $\mathbf{M}(12, 384) \rightarrow \mathbf{M}(12, 768)$ | | | | | | | | | | |
| bert2BERT | 5.7 | 35.6% | 92.89(0.65) | 84.92(0.19) | 86.91(0.70) | 60.32(2.16) | 91.81(0.34) | 88.11(0.57) | 90.72(0.13) | 85.10(0.68) |
| LiGO | 5.9 | 33.5% | 92.75(0.37) | 84.99(0.12) | 87.44(1.13) | 61.10(1.03) | 91.33(0.23) | 87.94(0.44) | 90.42(0.13) | 85.14(0.49) |
| Mango | 5.4 | **39.2%** | 92.71(0.20) | 84.86(0.22) | 87.94(1.11) | 62.88(0.86) | 91.49(0.12) | 88.73(0.35) | 90.62(0.20) | 85.60(0.44) |

Table 8: Detailed downstream results on the SQuADv1.1 and SQuADv2.0 datasets.

| Model | FLOPs (× 1e19) | Ratio (Saving) | SQuADv1.1 | | SQuADv2.0 | | SQuAD Avg. | |
|---|---|---|---|---|---|---|---|---|
| | | | F1 | EM | F1 | EM | F1 | EM |
| *Training from Scratch* | | | | | | | | |
| Scratch | 8.9 | - | 89.21(0.16) | 82.07(0.30) | 77.90(0.36) | 74.85(0.40) | 83.56(0.26) | 78.46(0.35) |
| StackBERT | 6.3 | 29.5% | 89.82(0.14) | 82.89(0.16) | 78.21(0.38) | 75.18(0.44) | 84.01(0.26) | 79.03(0.30) |
| *Training from the Pretrained Model:* $\mathbf{M}(12, 384) \rightarrow \mathbf{M}(12, 768)$ | | | | | | | | |
| bert2BERT | 5.7 | 35.6% | 90.02(0.36) | 83.24(0.52) | 78.99(0.31) | 75.98(0.35) | 84.50(0.34) | 79.61(0.43) |
| LiGO | 5.9 | 33.5% | 90.09(0.31) | 82.77(0.19) | 78.34(0.24) | 75.31(0.24) | 84.22(0.28) | 79.04(0.21) |
| Mango | 5.4 | **39.2%** | 90.17(0.27) | 83.29(0.33) | 78.77(0.18) | 75.71(0.14) | 84.47(0.22) | 79.50(0.24) |

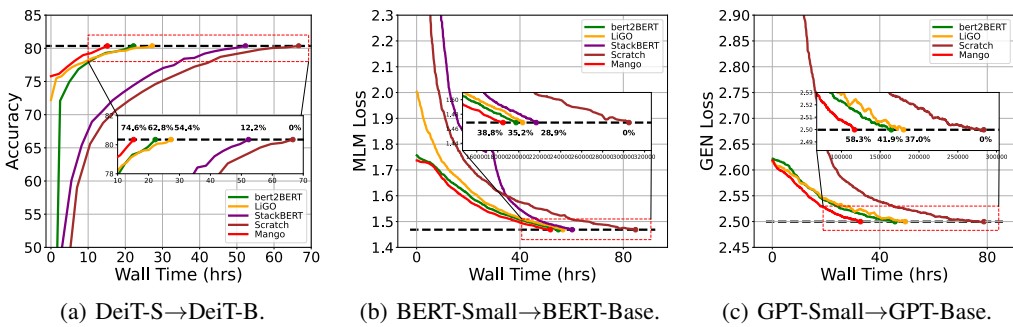

(a) DeiT-S→DeiT-B.   (b) BERT-Small→BERT-Base.   (c) GPT-Small→GPT-Base.

Figure 10: Results of pretraining DeiT-B, BRET-Base and GPT-Base on the wall time.