# OpenReview forum: "Reusing Pretrained Models by Multi-linear Operators for Efficient Training"
_NeurIPS.cc/2023/Conference — NeurIPS 2023 poster_

### Official Review · Reviewer_cdCX · 2023-06-08

**Soundness:** 3 good
**Presentation:** 3 good
**Contribution:** 3 good
**Rating:** 6
**Confidence:** 3

**Summary:**

The authors propose a new approach, Mango, to use a small model as initializer to larger models in order to speed up the training of the larger model. Mango proposes to learn a mapping of weights from the smaller model to the larger model, not layer by layer or other restricted form of mapping, but a full mapping of weights.

Because the matrix that would represent this linear mapping of weights being huge (in the order of M x N where M and N are respectively the number of parameters of the small and the large model), Mango proposes a particular form of low rank factorization, allowing to keep a full mapping but constraining each of its sub-mapping to rank 1. The authors show that this mapping can be learned in as few as 100 steps, making its computing cost negligible.

By doing so, Mango shows that full mappings are useful, and shows faster convergence than the other methods on 3 types of architectures : vision transformers, BERT models and GPT models.

**Strengths:**

* Well written paper, with very good motivating examples
* Good theoretical part, with the need of low rank factorization clearly introduced
* Compelling results of 3 different models: Vision Transformers, BERT and GPT
* The gaps in the results are significant: on vision transformers, Mango converge in 2/3 of the time of the next best method, and in 1/4 of the time of naive from scratch training. The savings on BERT are less important but still quite significant.


**Weaknesses:**

The results are constrained to the case of Small to Base size transformers mapping :
* Deit-S to Deit-B
* BERT-S to BERT-M
* GPT-Base from GPT-Small

The most interesting regime for this class of method is with bigger transformers since they are the most power hungry. The paper would gain a lot of impact by showing that Mango works for larger models (for instance to initialize a ViT-G from a ViT-H). I would definitely increase my rating if such a proof was shown, especially with a low rank such as 1.

Another interesting result would be to know, as we deal with larger models, whether it is best to initialize them with the larger pre-trained available, or if we can initialize them with Mango from a small model. For example, to initialize a ViT-G, should we preferentially use a ViT-H or ViT-L ? And how much would the difference be ?

**Questions:**

* The authors mention the CO2 consumption of training large models. Do you have an estimate of those emissions and the type of reduction achieved by Mango? Is the improvement proportional to the improvement in FLOPs?

---

> ### Author Rebuttal · Authors · 2023-08-10
>
> We appreciate the reviewer's valuable comments!
>
> > The most interesting regime for this class of method is with bigger transformers since they are the most power hungry. The paper would gain a lot of impact by showing that Mango works for larger models (for instance to initialize a ViT-G from a ViT-H). I would definitely increase my rating if such a proof was shown, especially with a low rank such as 1.
> >
> > Another interesting result would be to know, as we deal with larger models, whether it is best to initialize them with the larger pre-trained available, or if we can initialize them with Mango from a small model. For example, to initialize a ViT-G, should we preferentially use a ViT-H or ViT-L ? And how much would the difference be?
>
> We construct an experiment to train a large model ViT-G-1B. We plot the early result in the uploaded figure pdf due to the short rebuttal period. Although the scratch model has not reached a high position, the tendency shows our acceleration on it. The results also suggest a larger model will have more benefits for Mango.
>
> > The authors mention the CO2 consumption of training large models. Do you have an estimate of those emissions and the type of reduction achieved by Mango? Is the improvement proportional to the improvement in FLOPs?
>
> Yes. The improvement is proportional to the improvement in FLOPs accoring to [1].
>
> [1] David A. Patterson, Joseph Gonzalez, Quoc V. Le, Chen Liang, Lluis-Miquel Munguia, Daniel Rothchild, David R. So, Maud Texier, Jeff Dean: Carbon Emissions and Large Neural Network Training. CoRR abs/2104.10350 (2021)

---

### Official Review · Reviewer_9N2L · 2023-06-30

**Soundness:** 3 good
**Presentation:** 3 good
**Contribution:** 3 good
**Rating:** 5
**Confidence:** 4

**Summary:**

This paper addresses the issue of high resource costs in training large AI models from scratch. To this end, the authors propose a novel approach that linearly correlates each weight of the larger target model to all weights of the smaller pre-trained model, employing multi-linear operators to decrease complexity and manage resource demands. Experimental results on ImageNet reveal that the proposed method can significantly reduce training costs while maintaining performance.

**Strengths:**

The main idea is well-written and easy to follow. The introduced Mango Operator is both innovative and intuitive, exhibiting a remarkable ability to reduce training costs.

**Weaknesses:**

The experimental section lacks essential details, particularly around how the acceleration ratio is measured. Also, there are potentially contradictory statements about training steps and epochs, and the use of only theoretical metrics such as FLOPs for measuring training cost.

**Questions:**

1.	In the proposed method, the authors first concatenate weights across layers. However, given that the row count of $\mathbf{W}_j^{OUT} \in kI \times O$ differs from that of other weight matrices, how is this discrepancy addressed? Elaborating on this point would enhance the clarity of the method.

2.	In Equation (5), the authors suggest decomposing $\mathbf{S}$ using a Tensor Ring Matrix Product Operator (TP-MPO) into four smaller tensors. I'm curious to know if it would be possible to decompose $\mathbf{S}$ into five or six smaller tensors.

3.	Essential details seem to be lacking in the experimental section. Could the authors clarify how the acceleration ratio was measured? It would be better to provide more descriptions, which would strengthen the paper.

4.	There are a few statements in the experiments section that cause confusion. The authors mention that the Mango operators are trained for 100 steps, which is claimed to take negligible time. However, they also state that the model is trained for 300 epochs. Could the authors clarify these seemingly contradictory statements?

5.	In the experiments, the authors measure the training cost using FLOPs, which is a theoretical metric. It might be more insightful if the authors could also offer measurements related to GPU training hours, providing a more practical perspective on computational cost.

6.	The proposed method significantly reduces training costs compared to the traditional 'training from scratch' approach, primarily by leveraging the knowledge of the pre-trained model. I am intrigued to understand if this approach would sustain its comparable performance if we were to remove the training cost restriction and instead train until convergence.


**Limitations:**

The authors have provided a dedicated discussion on the potential limitations of their work.

---

> ### Author Rebuttal · Authors · 2023-08-10
>
> We appreciate the thorough feedback provided by the reviewer!
>
> > In the proposed method, the authors first concatenate weights across layers. However, given that the row count of $\mathbf{W}_j^{OUT}\in \mathbb{R}^{kI\times O}$ differs from that of other weight matrices, how is this discrepancy addressed? Elaborating on this point would enhance the clarity of the method.
>
> Thanks for the suggestion. In the implement of Mango, we permute $\mathbf{W}_j^{OUT}\in \mathbb{R}^{kI\times O}$ to the shape of $I\times kO$. We will revise Figure 4 to fix this confusion.
>
> > In Equation (5), the authors suggest decomposing $\mathbf{S}$ using a Tensor Ring Matrix Product Operator (TP-MPO) into four smaller tensors. I'm curious to know if it would be possible to decompose $\mathbf{S}$ into five or six smaller tensors.
>
> Yes, it is possible to decompose $\mathbf{S}$ into five, six even more smaller tensors. The reason for decomposing $\mathbf{S}$ into four smaller tensors is to interpret it as the transformation of the four modes of the smaller models, as explained in Line 183. To be specific, $\mathbf{S}_B$ can capture the interactions among weights in the same layer. $\mathbf{S}_I$ and $\mathbf{S}_O$ denote the transformations of input and output dimensions, respectively. $\mathbf{S}_L$ indicates the relationship among layers. $R$ means the low-rank level of $\mathbf{S}$. Smaller $R$ means less correlation of these four tensors. Therefore, Mango can correlate all the weights in the small model, thereby achieving the full-mapping transformation, which demonstrates the effectiveness in training efficiency.
>
> >  The experimental section lacks essential details, particularly around how the acceleration ratio is measured.
> >
> > Essential details seem to be lacking in the experimental section. Could the authors clarify how the acceleration ratio was measured? It would be better to provide more descriptions, which would strengthen the paper.
>
> Thanks for your suggestion. We calculate the acceleration ratio items of the FLOPs. For example, assuming training from scratch converge to metric (accuracy or loss) value $\psi$ with FLOPs $a$, if Mango reach $\psi$ with FLOPs $b$, then the acceleration ratio can be calculated as
> $$
> 1-\frac{b}{a}.
> $$
> We will add a formulation to clarify the calculation of the acceleration ratio.
>
> > Also, there are potentially contradictory statements about training steps and epochs
> >
> > There are a few statements in the experiments section that cause confusion. The authors mention that the Mango operators are trained for 100 steps, which is claimed to take negligible time. However, they also state that the model is trained for 300 epochs. Could the authors clarify these seemingly contradictory statements?
>
> Sorry for the confusion. There are not contradictory statements for the training steps. We write the completed procedures of Mango at Line 194 of the paper. Specifically, the implement of Mango is divided into two phases. (1) Training Mango operators. We train the Mango operators (i.e., $\mathbf{S}_B$, $\mathbf{S}_I$, $\mathbf{S}_O$, and $\mathbf{S}_L$) for 100 steps for transfer the knowledge from the smaller pretrained model to the target model. This training cost is relatively small (usually several minutes), which can be negligible in comparison with the overall training of large models; (2) Training the target model. We derive the weight $W_2$ of the target model through Eq. (5.) with the trained Mango operators. Then, we train the target model initialized by $W_2$ for 300 epochs.
>
> >  the use of only theoretical metrics such as FLOPs for measuring training cost.
> >
> > In the experiments, the authors measure the training cost using FLOPs, which is a theoretical metric. It might be more insightful if the authors could also offer measurements related to GPU training hours, providing a more practical perspective on computational cost.
>
> Thanks for the suggestion. We upload the wall time pdf.
>
> > The proposed method significantly reduces training costs compared to the traditional 'training from scratch' approach, primarily by leveraging the knowledge of the pre-trained model. I am intrigued to understand if this approach would sustain its comparable performance if we were to remove the training cost restriction and instead train until convergence.
>
> When removing the training cost restriction, Mango can achieve better performance at the end of training. For example, in the experiment of "DeiT-S -> DeiT-B", Mango can improve the accuracy from 80.45% to 80.75% in the end. Similarly, Mango also achieves loss reduction as "BERT-Small -> BERT-Base": 1.47->1.41, "GPT-Small -> GPT-Base": 2.49->2.46.

---

> > ### Comment · Reviewer_9N2L · 2023-08-15
> > **Response to Authors**
> >
> > Dear Authors,
> >
> > Thanks for the reviewers' rebuttal, which has solved all of my concerns.
> >
> > Warm regards,
> >
> > Reviewer 9N2L

---

> > > ### Author Response · Authors · 2023-08-18
> > > **Response to Reviewer 9N2L**
> > >
> > > Dear Reviewer 9N2L,
> > >
> > > Thanks for your thoughtful engagement with the manuscript! We are pleased to solve all of your concerns. We appreciate your valuable comments for refining our work!
> > >
> > > Best regards,
> > >
> > > Authors

---

### Official Review · Reviewer_ShVL · 2023-07-01

**Soundness:** 3 good
**Presentation:** 3 good
**Contribution:** 3 good
**Rating:** 6
**Confidence:** 2

**Summary:**

This paper proposes a new method for reusing pre-trained models to a larger target model. They propose mango which decomposes the parameters into 4 smaller matrices and then learns to grow these parameters. They show their method requires less FLOPS to achieve as good as performance other baselines across vision and NLP tasks.

**Strengths:**

The paper was easy to follow and the experiments are strong across various domains.
Mango seems well motivated in allowing for interactions between all weights in the model.


**Weaknesses:**

Missing background on tensor ring matrix product - since this is the main method proposed
Assuming the acceleration ratio is in terms of FLOPS, then there are no experiments showing the wall-clock time of Mango. Decreasing FLOPS is only useful if it decreases wall-clock time (assuming the same machine).
The models seem to be small, only growing from small to base variants, not base to large variants


**Questions:**

Is the acceleration in terms of FLOPS or wall clock time?
How does the multiring operator allow for interactions between all weights in the model?
How does S_L capture the relationship between layers?


**Limitations:**

See weaknesses above.

---

> ### Author Rebuttal · Authors · 2023-08-10
>
> We appreciate the reviewer for the meaningful comments!
>
> > Missing background on tensor ring matrix product - since this is the main method proposed
>
> Thanks for the suggestion. We will add the background of TR-MPO in the "Preliminary" section.
>
> > Is the acceleration in terms of FLOPS or wall clock time?
> >
> > Assuming the acceleration ratio is in terms of FLOPS, then there are no experiments showing the wall-clock time of Mango. Decreasing FLOPS is only useful if it decreases wall-clock time (assuming the same machine).
>
> Yes, we calculate the acceleration ratio in terms of FLOPS. We also uploaded the wall time pdf.
>
> > The models seem to be small, only growing from small to base variants, not base to large variants
>
> We construct an experiment to train a large model ViT-G-1B. We plot the early result in the uploaded figure pdf due to the short rebuttal period. Although the scratch model has not reached a high position, the tendency shows our acceleration on it.
>
> > How does the multiring operator allow for interactions between all weights in the model?
> >
> > How does S_L capture the relationship between layers?
>
> As shown in Equation (5), each element within the target model $M_2$ results from the summation of the product between trainable weights and each weight from the small model $M_1$, thereby achieveing interactions between all weights in the model.
>
> Additionally, Mango assigns trainable weights (denoted as $\mathbf{S}_L$) to facilitate the learning of layer importance within smaller models.

---

### Official Review · Reviewer_kqva · 2023-07-10

**Soundness:** 3 good
**Presentation:** 3 good
**Contribution:** 3 good
**Rating:** 5
**Confidence:** 5

**Summary:**

This paper studied the problem of solving a proper initialization for large deep learning models on top of smaller pre-trained networks. The key idea is to concatenate the weights of pre-trained models to a large matrix, and map the matrix to the parameter space of the larger model to be trained with a multi-linear operator. In this procedure, the rank of the transformation matrix can be manually adjusted. Experimental results on NLP/CV-based Transformer models are provided.

**Strengths:**

1. The problem of efficient training is practically important.
2. The proposed method is well-motivated.
3. The design of the multi-linear operator is interesting.
4. Mango can be applied to both NLP and CV models.

**Weaknesses:**

1. The writing needs to be improved. It is difficult to understand Eqs. (4-5), which are the key contributions of this paper.
2. Currently, the topology of the network is not considered in Mango (e.g., the depth of the layer corresponding to each weight matrix). I think it may be beneficial to leverage this characteristic in Mango.
3. Can we introduce non-linear operation to Mango to improve training efficiency?
4. In Figure 7, the comparison with training from scratch may be unfair. Take DeiT for example. Obtaining DeiT-S needs a notable amount of computation. Hence, the computational cost of Mango should not start from zero FLOPs.
5. This basic baseline may be interesting: initializing the large model (e.g., DeiT-B) by pre-training it using the same amount of computation as training the smaller model (e.g., DeiT-S).
6. Can Mango improve the final accuracy of the model?
7. In the field of CV, it would be interesting to validate the effectiveness of Mango on top of more network architectures, e.g., Swin Transformer.
8. Some related works should be discussed or compared with Mango:
- Li, C., Zhuang, B., Wang, G., Liang, X., Chang, X., & Yang, Y. (2022). Automated progressive learning for efficient training of vision transformers. In Proceedings of the IEEE/CVF Conference on Computer Vision and Pattern Recognition (pp. 12486-12496).
- Budgeted Training for Vision Transformer, ICLR'23
- Wang, Y., Yue, Y., Lu, R., Liu, T., Zhong, Z., Song, S., & Huang, G. (2022). Efficienttrain: Exploring generalized curriculum learning for training visual backbones. arXiv preprint arXiv:2211.09703.

**Questions:**

Please see the weaknesses.

**Limitations:**

Personally, I think this paper may not have potential negative societal impacts.

---

> ### Author Rebuttal · Authors · 2023-08-10
>
> We thank the reviewer for the thoughtful comments!
>
> > The writing needs to be improved. It is difficult to understand Eqs. (4-5), which are the key contributions of this paper.
>
> Eqs. (4-5) are both sum of element-wise production among tensors. The subscripts (e.g., $b_{\ast}, i_{\ast}, o_{\ast}, l_{\ast}, o_{\ast}$, and $r_{\ast}$) mean the entries of the tensors. We will try our best to further polish the clarification, e.g., adding more description of the tensors and calculation in the equations .
>
> > Currently, the topology of the network is not considered in Mango (e.g., the depth of the layer corresponding to each weight matrix). I think it may be beneficial to leverage this characteristic in Mango.
>
> This is a great comment and could be a future perspective to explore. If our understanding of the term "topology" is accurate, Mango may incorporate topology-related information into its considerations. Considering the example of layer topology to illustrate this concept, it is noteworthy that different layers typically have distinct functionality. Therefore, accounting for these variations can be beneficial. An interesting aspect is that Mango assigns trainable weights (denoted as $\mathbf{S}_L$) to facilitate the learning of layer importance within smaller models. This approach implicitly captures the aforementioned topology in some sense. Further utilization of network topology is beyond the scope of this work and can be explored in the future.
>
> > Can we introduce non-linear operation to Mango to improve training efficiency?
>
> We apply a non-linear activation function GeLU after multiplying the smaller model with Mango operators. We construct this experiment by training DeiT-S from DeiT-T-A on ImageNet. The result shows GeLU can achieve acceleration ratios at 31.0% which is the same as training without a non-linear activation function. This preliminary exploration of non-linear activation functions shows that it is possible to apply non-linear functions. However, an efficient way to apply non-linear functions is still needed to be explored.
>
>
>
> > In Figure 7, the comparison with training from scratch may be unfair. Take DeiT for example. Obtaining DeiT-S needs a notable amount of computation. Hence, the computational cost of Mango should not start from zero FLOPs.
> >
> > This basic baseline may be interesting: initializing the large model (e.g., DeiT-B) by pre-training it using the same amount of computation as training the smaller model (e.g., DeiT-S).
>
> This is a good perspective. We understand that the cost of pretraining small models could be added into the total cost of the proposed method and prior research (e.g., bert2BERT and LiGO), for a fair comparison with training from scratch. However, it is important to restate the setting of this paper and prior research (e.g., bert2BERT and LiGO) is that there are free and publicly available small or medidum scaled pretrained models (e.g., those can be downloaded from HuggingFace or released by public giants), while the cost of training large models from scrach is prohibited. Given the existing smaller pretrained models, Mango effectively leverages the insights within smaller pretrained models, thereby can achieve significant improvement in training efficiency in comparison with  prior research (e.g., bert2BERT and LiGO) and training from scratch. Our experiments, based on standard model configurations, demonstrate the practicality of utilizing available pretrained models to expedite training. Furthermore, we believe that the total cost of training small models and the expanded model training should be much smaller than large models (especially for LLMs which are often computationally prohibitive). Consequently, **it is reasonable to make a comparison between methods of training for a smaller pretrained model (e.g., Mango, bert2BERT, and LiGO) and training from scratch.**
>
> Nonetheless, it is important to acknowledge that the necessity for a pretrained model remains a potential limitation, which we clearly address in the "Limitations" section. We hold the view that while this limitation exists, its impact on real-world applicability is not overly stringent.
>
>
> > Can Mango improve the final accuracy of the model?
>
> Yes. With follow-up training, Mango can achieve better performance. For example, in the experiment of "DeiT-S -> DeiT-B", Mango can improve the accuracy from 80.45% to 80.75% in the end. Similarly, Mango also achieves loss reduction as "BERT-Small -> BERT-Base": 1.47->1.41, "GPT-Small -> GPT-Base": 2.49->2.46.
>
> > In the field of CV, it would be interesting to validate the effectiveness of Mango on top of more network architectures, e.g., Swin Transformer.
>
> We conduct an experiment on training Swin-S. Mango adopts a Swin-T model as the smaller model. We train Swin-S for 240 epoch. We show a acceleration in the uploaded figure pdf.
>
>
> > Some related works should be discussed or compared with Mango:
>
> Thanks for the suggestions regarding the related works. We appreciate your insights and have carefully considered the references you provided. [1] employs a neural architecture search (NAS) method to search optimal sub-networks for progressive training on ViTs. [2]  suggests a progressive growing mechanism used across a three-phase training regime to achieve a good trade-off between training budget and performance. [3] introduces a novel curriculum learning approach for training efficiency through firstly learning simpler patterns, then progressively introducing more complex patterns. We will add these related studies to the "Related Work" section of the paper for comparison.

---

> > ### Comment · Reviewer_kqva · 2023-08-15
> > **Post-rebuttal comments**
> >
> > I highly appreciate the efforts of the authors in rebuttal. Most of my concerns have been addressed. I'm happy to keep my positive rating (i.e., Borderline Accept).

---

> > > ### Author Response · Authors · 2023-08-18
> > > **Response to the post-rebuttal comments**
> > >
> > > Thank you for your valuable feedback! We are glad to know that most of your concerns have been addressed to your satisfaction. Your comment has been immensely helpful in improving the quality of our work. We appreciate it much for your positive feedback.

---

### Official Review · Reviewer_u4ot · 2023-07-26

**Soundness:** 3 good
**Presentation:** 3 good
**Contribution:** 2 fair
**Rating:** 6
**Confidence:** 4

**Summary:**

The contribution of this paper lies in proposing a new method to accelerate the training of large models by leveraging the correlations among all weights in the model, instead of just mapping partial weights as in existing methods. The authors' approach utilizes multi-linear operators to reduce computational and spatial complexity, enabling acceptable resource requirements. Experimental results demonstrate that their method has excellent performance in accelerating training, outperforming existing methods.

**Strengths:**

+ The method proposed in the paper can leverage the potential correlations among all weights in the model, significantly reducing computational costs.
+  the authors conducted experiments in both image and natural language processing domains and achieved results that surpass existing methods. This indicates that their proposed approach has a certain level of generality and applicability, and can significantly accelerate model training across different domains.

**Weaknesses:**

+ In the experiments of the paper, only DeiT-B, BERT-Base, and GPT-Base were used, and no further exploration was done on larger models.
+ The performance of Mango, Ligo, and bert2BERT was improved in natural language processing methods, while only marginal gains were achieved in image tasks, and the experiment lacked analysis and explanations.
+ For Figure 6(b), there is continuous improvement in accuracy with increasing rank, but there is a lack of experimental comparison for larger rank values.

**Questions:**

+ In line 196 of the paper, for the loss function in Equation (6), what is the specific design of this loss function for the operators? Is it task-specific or fixed?

**Limitations:**

It might be interesting if the results can be extended to large language models.

---

> ### Author Rebuttal · Authors · 2023-08-10
>
> We are thankful for the insightful comments of the reviewer!
>
> > In the experiments of the paper, only DeiT-B, BERT-Base, and GPT-Base were used, and no further exploration was done on larger models.
>
> We construct an experiment to train a large model ViT-G-1B. We plot the early result in the uploaded figure pdf due to the short rebuttal period. Although the scratch model has not reached a high position, the tendency shows our acceleration on it.
>
> > The performance of Mango, Ligo, and bert2BERT was improved in natural language processing methods, while only marginal gains were achieved in image tasks, and the experiment lacked analysis and explanations.
>
> This is an interesting phenomenon. We construct downstream tasks with the checkpoint of the same metric value (accuracy in vision task and loss in language task). The similar result in vision downstream tasks suggests that reusing a small pretrained model for training efficiency will not hurt the transferring ability of the target model. By contrast, downstream tasks in language show resuing a pretrained model can make an improvement. We guess the trained target may cause this difference, vision: main task: classification accuracy -> downstream task: classification accuracy; language: main task: pretraining loss -> downstream task: classification accuracy. However, as deep networks are in a black box, we cannot precisely interpret this phenomenon. In all, this phenomenon suggests that the methods (e.g., Mango, bert2BERT and LiGO) reusing a small pretrained model can improve the training efficiency without hurting model transferring ability, and derive a performance profit in downstream tasks mainly in language models when the loss is comparable.
>
> > For Figure 6(b), there is continuous improvement in accuracy with increasing rank, but there is a lack of experimental comparison for larger rank values.
>
> We add a rank 24 to the experiment of "DeiT-T-B -> DeiT-S". Results are shown in the following table. The accuracy of the larger rank of 24 is slightly less than the rank of 10, while still achieving the same acceleration ratio as ranks of 4, 7, and 10. Therefore, as we also suggest in the Line 253 of the paper, rank 1 is enough to use, and also enjoys the advantages of spatial and temporal complexities compared to bigger ranks.
>
> |                    |   1    |   4    |   7    |   10   |   24   |
> |:------------------ |:------:|:------:|:------:|:------:|:------:|
> | Accuracy           | 73.72% | 73.75% | 73.76% | 73.81% | 73.77% |
> | Acceleration Ratio | 46.0%  | 45.7%  | 45.7%  | 45.7%  | 45.7%  |
>
>
> > In line 196 of the paper, for the loss function in Equation (6), what is the specific design of this loss function for the operators? Is it task-specific or fixed?
>
> The loss function is consistent to the training of the target model. For example, when we train the target model in CV tasks with a classification loss, we adopt the same classification loss for training the operators. Similarly, we use the MLM loss in BERT models and the GEN loss in GPT models for training corresponding Mango operators.

---

> > ### Comment · Reviewer_u4ot · 2023-08-15
> >
> > I highly appreciate the efforts of the authors in rebuttal. Most of my concerns have been addressed.

---

> > > ### Author Response · Authors · 2023-08-18
> > >
> > > Thank you for recognizing our work! We are delighted to hear that most of your concerns have been addressed to your satisfaction. We sincerely appreciate your thoughtful review and are grateful for the opportunity to enhance our research based on your insights.

---

### Author Rebuttal · Authors · 2023-08-10

We deeply appreciate the insightful and considerate feedback provided by all of the reviewers.  We are glad that the reviewer kqva notes that efficient training is practically important, and all the reviewers (u4ot, kqva, ShVL, 9N2L, and cdCX) consider the proposed Mango can achieve significant acceleration on the training process. We are also appreciative of the reviewers (kqva, ShVL, 9N2L, and cdCX) who consider that Mango is well-motivated, novel, and interesting. Moreover, we also thank reviewers (9N2L and cdCX) who commend the clarity and accessibility of our writing. In addition, we have made earnest endeavors to thoroughly address all the inquiries raised by the reviewers, encompassing additional experimental comparisons and clarifications. The extra empirical results have been plotted and are readily accessible within the uploaded PDF document.

---

### Decision · Program_Chairs · 2023-09-21

**Decision:**

Accept (poster)

**Comment:**

This paper proposes an approach reducing the required training time for a model by reusing the weights of a smaller network.  Through the course of the reviewing process the authors and reviewers discussed the merits of the proposed approach, with all reviewers recommending acceptance.  The ACs find that there is not sufficient justification to overturn the unanimous recommendation of the reviewers, and remind the authors to use the comments and rebuttals to update their paper given the reviewer's feedback.